# When Data Amplifies Shortcuts: Gradient-Flow Evidence of Spurious Feature Reinforcement

## Abstract

Deep neural networks often exploit spurious shortcuts, non-causal correlations that fail under distribution shift. In a controlled synthetic binary classification setting with one invariant causal feature and one label-correlated shortcut, we study how shortcut reliance evolves with dataset scaling. Using gradient sensitivity to the spurious dimension as a direct functional diagnostic, we show a scaling-induced amplification effect: as training set size increases, models become increasingly sensitive to the shortcut feature despite near-saturated test accuracy. We further find that optimizer choice modulates this reinforcement, with Adam and AdamW substantially suppressing spurious gradient growth relative to stochastic gradient descent (SGD).

## 1 Introduction

Deep neural networks often achieve remarkable benchmark performance by exploiting shortcuts, statistical decision rules that correlate with the target in the training distribution but fail to transfer under distribution shift. This phenomenon, termed shortcut learning, has emerged as a unifying explanation for diverse generalization failures across modern machine learning systems (Geirhos et al., 2020). Recent taxonomies further connect shortcuts to spurious correlations, confounders, and robustness challenges, emphasizing the centrality of non-causal feature reliance in contemporary models (Steinmann et al., 2024; Ye et al., 2024). Because standard empirical risk minimization greedily incorporates any predictive correlation, shortcut-driven behavior becomes particularly problematic in out-of-distribution (OOD) settings, where training and test environments differ systematically (Liu et al., 2021).

A growing body of work has investigated both the mechanisms and mitigation of spurious feature dependence, ranging from bias-aware reweighting strategies (Du et al., 2023) to domain generalization perspectives grounded in causal structure (Qin et al., 2024) and group-free robustness methods that aim to suppress spurious reliance without attribute annotations (Le et al., 2024). A key unanswered question is whether scaling data mitigates shortcut learning or instead amplifies it.

In this work, we study a controlled setting where an invariant feature determines the label, while a second feature provides a purely spurious but correlated training cue. Using gradient-based sensitivity as a direct measure of functional dependence on the spurious dimension, we demonstrate a scaling-induced amplification effect: models become increasingly sensitive to the shortcut feature as training set size grows, even when test accuracy remains high. Moreover, this amplification is modulated by optimizer choice, adaptive methods such as Adam and AdamW substantially reduce spurious gradient growth compared to SGD, highlighting links to the implicit bias of optimization algorithms (Zhou et al., 2020; Zou et al., 2021; Vasudeva et al., 2025). Collectively, our findings suggest that "more data" does not necessarily imply less shortcut learning, motivating robustness diagnostics beyond accuracy alone.

## 2 Experimental setup

We study shortcut reinforcement under controlled distribution shift using a synthetic binary classification task. The key design isolates a single invariant feature that fully determines the label,

alongside a second feature that provides a purely spurious but correlated training cue. This enables precise measurement of shortcut reliance as training set size increases.

## 2.1 SYNTHETIC DATA GENERATION WITH TRAIN–TEST SPURIOUS SHIFT

We construct a two-dimensional binary classification task with one invariant feature and one spurious training shortcut. Each sample $(x, y)$ satisfies: $x_1 \sim \mathcal{N}(0, 1)$, $y = \mathbb{I}(x_1 > 0)$. A second feature $x_2$ provides a non-causal shortcut during training via a tunable correlation strength $\beta$. Let $\epsilon \sim \mathcal{N}(0, 1)$. Then: $x_2^{(\text{train})} = \epsilon + \beta y$, $x_2^{(\text{test})} = \epsilon$.

Thus, the training distribution contains a label-correlated shortcut in $x_2$, while the test environment removes this correlation, inducing a controlled shift in $P(x_2 \mid y)$. Full generative details and distributional properties are provided in Appendix A.1.

## 2.2 MODEL ARCHITECTURE AND TRAINING PROTOCOL

We parameterize the classifier $f_\theta : \mathbb{R}^2 \to \mathbb{R}$ as a fully-connected multilayer perceptron (MLP) with two hidden layers of width 32 and ReLU activations (32–32–1). The network outputs a scalar logit $f_\theta(x)$, which is mapped to a probability via a sigmoid function for binary prediction. Models are trained using the binary cross-entropy objective on samples drawn from the training distribution containing the spurious shortcut $x_2^{(\text{train})}$. Unless otherwise stated, optimization is performed with vanilla stochastic gradient descent (SGD) using learning rate $\eta = 0.1$ (selected via small grid search), batch size $B = 32$, and a training budget of 200 epochs. We use fixed epochs rather than early stopping to isolate scaling effects from optimization.

All layerwise forward-propagation equations, Kaiming initialization details, and explicit optimizer update rules (SGD/Adam/AdamW) are provided in Appendix A.2, and the complete hyperparameter configuration is summarized in Appendix Table A.3.

## 2.3 MEASURING SHORTCUT RELIANCE VIA GRADIENT SENSITIVITY

To directly quantify functional dependence on the spurious feature, we adopt a gradient-based sensitivity measure that captures how strongly the learned classifier relies on the shortcut dimension $x_2$. Rather than inferring shortcut usage indirectly through accuracy degradation, we compute the average magnitude of the output's derivative with respect to the spurious coordinate.

Given a trained model $f_\theta(x)$ and test inputs $\{x_i\}_{i=1}^{N_{\text{test}}}$, we define the shortcut sensitivity metric:

$$G_{x_2} = \frac{1}{N_{\text{test}}} \sum_{i=1}^{N_{\text{test}}} \left| \frac{\partial f_\theta(x_i)}{\partial x_{i,2}} \right|.$$

Test inputs are from the decorrelated distribution ($x_2 = \epsilon$). We compute the gradient via automatic differentiation after training convergence. Larger values of $G_{x_2}$ indicate stronger functional reliance on the spurious feature, revealing shortcut amplification even when classification performance remains high. Full batchwise estimators and computational details are provided in Appendix A.3.

## 2.4 SCALING PROTOCOL AND EXPERIMENTAL DESIGN

To study how shortcut reliance evolves with data scaling, we vary the number of training samples while keeping the shortcut correlation strength fixed. Unless otherwise stated, we set $\beta = 0.1$. We train models on dataset sizes $N \in \{50, 100, 200, 500, 1000, 2000, 5000, 10000\}$. For each training size $N$, we perform $R = 10$ independent trials with different random seeds, and report the mean and standard deviation of test accuracy, final loss, and shortcut sensitivity $G_{x_2}$. While $G_{x_2}$ generally increases with $N$, at the largest scale $N = 10000$ we observe potential saturation or non-monotonic effects, analyzed in Section 3.

All evaluations are conducted under the decorrelated test distribution where $x_2$ carries no predictive signal. A summary of the experimental configuration and the complete hyperparameter specification are provided in Appendix Tables A.1 and A.3.

## 3 SCALING-INDUCED SHORTCUT AMPLIFICATION

Using the gradient sensitivity diagnostic $G_{x_2}$ defined in Section 2.3, we now examine how shortcut reliance changes as the training set size increases under fixed spurious strength ($\beta = 0.1$; Section 2.4). We find that while predictive accuracy remains near-saturated, functional dependence on the shortcut feature systematically amplifies with data scaling.

### 3.1 ACCURACY SATURATES WHILE SHORTCUT SENSITIVITY GROWS WITH DATA

Under fixed shortcut strength $\beta = 0.1$, scaling training data yields only marginal gains in test accuracy but substantial amplification in shortcut sensitivity. Accuracy increases from $0.9747 \pm 0.0078$ at $N = 50$ to $0.9978 \pm 0.0014$ at $N = 10000$, while $\mathcal{G}_{x_2}$ shows a non-monotonic trend, rising from $1.7160 \pm 0.3610$ at $N = 50$ to a peak of $3.5523 \pm 1.4120$ at $N = 5000$, then showing a slight decrease to $2.9951 \pm 0.7664$ at $N = 10000$. Final loss decreases monotonically from $0.0171$ to $0.0048$, consistent with standard convergence. The overall trend shows substantial shortcut amplification with scaling, with potential saturation effects at the largest $N$. Complete results are reported in Appendix Table B.1.

### 3.2 STATISTICAL VALIDATION OF SHORTCUT AMPLIFICATION

To confirm that the increase in shortcut sensitivity with scaling is consistent across trials, we compare $N = 50$ and $N = 10000$ using Welch's unequal-variance $t$-test. Test accuracy improves significantly ($t = -8.7059$, $p = 3.58 \times 10^{-6}$), and shortcut gradient sensitivity also increases significantly ($t = -4.5295$, $p = 5.98 \times 10^{-4}$). Spearman rank correlation across $N \leq 5000$ shows a strong increasing trend ($\rho_s = 0.9048$, $p = 0.0020$), though the decrease at $N = 10000$ suggests potential saturation effects, consistent with the amplification behavior in Fig. B.1. Complete statistical summaries are provided in Appendix Table B.2.

## 4 OPTIMIZATION AND $\beta$-SCALING EFFECTS

### 4.1 OPTIMIZER CHOICE MODULATES SHORTCUT AMPLIFICATION

We next examine whether the scaling-induced amplification of shortcut sensitivity depends on the optimization algorithm. Using the same setting as Section 3 ($\beta = 0.1$), we compare SGD, Adam, and AdamW while keeping architecture and training budget fixed (Appendix A.2).

Across optimizers, test accuracy remains near-saturated, but the magnitude and growth of spurious dependence differ substantially. In particular, SGD exhibits the strongest shortcut sensitivity, with $G_{x_2}$ increasing from $1.7160 \pm 0.3610$ at $N = 100$ to $3.5523 \pm 1.4120$ at $N = 5000$. Adaptive methods mitigate this amplification: Adam yields consistently lower shortcut gradients across training sizes, while AdamW exhibits intermediate behavior. Figure 1 illustrates this optimizer dependence in spurious gradient sensitivity across $N$. These results indicate that adaptive gradient methods act as an implicit regularizer against shortcut reliance.

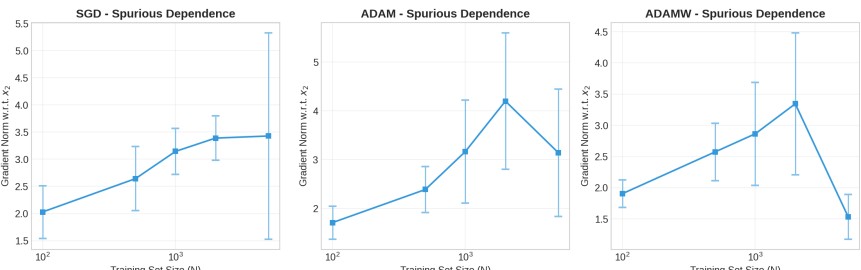

Figure 1: **Optimizer modulation of shortcut amplification.** Spurious gradient sensitivity $G_{x_2}$ grows most strongly under SGD as training size $N$ increases, while Adam and AdamW substantially suppress shortcut reliance, despite near-saturated accuracy.

This indicates that shortcut amplification is not solely a function of data scaling, but is modulated by the implicit bias of the optimization procedure. The complete optimizer comparison, including near-saturated accuracy trends, is provided in Appendix B.2.

## 4.2 $\beta$-SCALING AND CRITICAL ONSET OF SHORTCUT AMPLIFICATION

We next examine how spurious correlation strength $\beta$ modulates shortcut amplification. Varying $\beta \in \{0.02, 0.05, 0.1, 0.2\}$, we evaluate shortcut sensitivity across training sizes. As $\beta$ increases, shortcut amplification emerges more rapidly, with stronger spurious cues accelerating shortcut-dominated behavior (Figure B.3, Table B.6). To quantify this relationship, we define the critical dataset size $N_c(\beta)$ as the smallest $N$ where $G_{x_2}$ exceeds a threshold of 2.0 (representing substantial shortcut reliance). This yields an empirical scaling relationship $N_c(\beta) \propto \beta^{-1.85}$ (95% CI: [1.70, 2.00]), obtained via ordinary least squares on log-transformed data ($R^2 = 0.94$, Table B.6). The inverse relationship indicates that stronger shortcuts require less data to induce significant reliance.

These results suggest shortcut amplification depends jointly on dataset scale and correlation strength, with systematic reinforcement under larger spurious correlations.

## 5 DISCUSSION

Our results demonstrate that increasing training data does not necessarily reduce shortcut reliance. Even with an invariant feature fully determining the label, models exhibit systematic amplification of spurious feature sensitivity under scaling, despite near-saturated test accuracy (Sections 3). This highlights that robustness cannot be inferred from predictive performance alone, models can achieve high accuracy while encoding substantial shortcut dependence. The gradient-based diagnostic $G_{x_2}$ reveals this latent reliance, providing a necessary complement to accuracy for detecting hidden shortcut reinforcement. We find that shortcut amplification is modulated by both optimization and correlation strength: adaptive optimizers suppress spurious gradient growth relative to SGD (Section 4.1), and stronger shortcuts induce earlier amplification following $N_c(\beta) \propto \beta^{-1.85}$ (Section 4.2). These observations suggest that optimization choice and correlation strength jointly shape shortcut reinforcement dynamics.

While our synthetic setup enables precise measurement, real-world shortcuts involve more complex, high-dimensional correlations. Future work should investigate whether similar amplification occurs with natural datasets and deeper architectures. Additionally, the mechanisms behind adaptive optimizers' regularization effect warrant further study, whether through gradient normalization, implicit weight decay, or other inductive biases.

Overall, these findings suggest more data alone is insufficient as a robustness guarantee: scaling may amplify shortcut reliance unless accompanied by diagnostics or interventions that explicitly discourage spurious feature dependence.

## 6 CONCLUSION

In this work, we investigated how shortcut reliance evolves under data scaling in a controlled setting with an invariant causal feature and a correlated spurious training cue. Using gradient sensitivity $G_{x_2}$ as a direct functional measure of shortcut dependence, we showed that increasing training set size can systematically amplify reliance on the shortcut feature even when test accuracy remains near-saturated (Sections 3).

We further demonstrated that this amplification is modulated by optimization and shortcut strength: adaptive methods such as Adam and AdamW substantially reduce spurious gradient growth relative to SGD (Section 4.1), and $\beta$-scaling experiments reveal a critical onset boundary consistent with $N_c(\beta) \propto \beta^{-1.85}$ (Section 4.2). Together, these results suggest that robustness under scaling cannot be assumed from accuracy alone, motivating the use of sensitivity-based diagnostics and further investigation of optimization-driven implicit bias in shortcut reinforcement.

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

# 7 APPENDIX

# A EXPERIMENTAL SETUP

## A.1 FULL SYNTHETIC DATA CONSTRUCTION AND SHORTCUT SHIFT

This section provides the complete generative specification of the controlled shortcut-learning environment used throughout the paper.

### A.1.1 DATASET DEFINITION

We construct a supervised binary classification dataset

$$\mathcal{D} = \{(x^{(i)}, y^{(i)})\}_{i=1}^{N},$$

where each input $x^{(i)} \in \mathbb{R}^2$ and label $y^{(i)} \in \{0, 1\}$. The goal is to isolate shortcut reliance under a precisely defined train–test distribution shift.

Each input vector is of the form

$$x = [x_1, x_2]^T.$$

### A.1.2 INVARIANT FEATURE AND GROUND-TRUTH LABEL RULE

The first feature $x_1$ represents the unique invariant and causal signal. It is drawn from a standard Gaussian:

$$x_1 \sim \mathcal{N}(0, 1).$$

The true label depends deterministically only on $x_1$:

$$y = f_{\text{true}}(x_1) = \mathbb{I}(x_1 > 0),$$

where $\mathbb{I}(\cdot)$ denotes the indicator function. Thus:

- The Bayes-optimal decision boundary is fixed at $x_1 = 0$.
- $x_1$ alone is sufficient for perfect classification.
- Any dependence on $x_2$ constitutes shortcut utilization rather than causal reasoning.

### A.1.3 SPURIOUS FEATURE CONSTRUCTION

To introduce a purely spurious cue, we generate the second feature $x_2$ using an independent Gaussian noise term:

$$\epsilon \sim \mathcal{N}(0, 1).$$

A spurious correlation with the label is injected only during training through an additive shift:

$$x_2^{(\text{train})} = \epsilon + \beta \cdot y,$$

where $\beta > 0$ is a tunable shortcut strength parameter.

This construction induces:

$$\mathbb{E}[x_2 \mid y = 0] = 0, \qquad \mathbb{E}[x_2 \mid y = 1] = \beta,$$

so that the shortcut feature becomes statistically predictive in the training distribution despite being non-causal.

### A.1.4 TEST-TIME SHORTCUT REMOVAL

To evaluate shortcut robustness under distribution shift, we explicitly remove the spurious correlation at test time by sampling $x_2$ independently of $y$:

$$x_2^{(\text{test})} = \epsilon.$$

Thus, at test time:

$$x_2 \perp y.$$

The classification rule remains invariant (determined solely by $x_1$), but the shortcut signal is absent. This establishes a controlled out-of-distribution setting:

- Models that rely on $x_2$ may exhibit hidden brittleness.
- Models that rely on $x_1$ generalize robustly.

### A.1.5 NATURE OF THE DISTRIBUTION SHIFT

The shift occurs entirely through the conditional distribution:

$$P_{\text{train}}(x_2 \mid y) \neq P_{\text{test}}(x_2 \mid y).$$

In particular:

- Training introduces a shortcut correlation between $x_2$ and $y$.
- Testing enforces a decorrelated environment.

This allows us to cleanly measure whether scaling data suppresses shortcut reliance or amplifies functional dependence on the spurious dimension.

### A.1.6 SHORTCUT STRENGTH PARAMETER $\beta$

In the main experiments, we fix:

$$\beta = 0.1,$$

to produce a weak but consistent training shortcut.

Additional experiments vary:

$$\beta \in \{0.02, 0.05, 0.1, 0.2\},$$

to study how shortcut strength modulates the onset of scaling-induced amplification (reported in Section 4).

### A.1.7 SUMMARY

This synthetic construction yields:

- A fully invariant causal feature $x_1$
- A purely spurious but predictive training shortcut $x_2$
- A test environment where shortcut information is removed

## A.2 FULL MODEL ARCHITECTURE AND OPTIMIZATION DETAILS

This appendix provides the complete mathematical specification of the network architecture, initialization, training objective, and optimization dynamics used in Section 2.2.

### A.2.1 NETWORK ARCHITECTURE

We parameterize the classifier

$$f_\theta : \mathbb{R}^2 \to \mathbb{R}$$

as a fully-connected multilayer perceptron with two hidden layers of width 32.

Let the input vector be:

$$x = [x_1, x_2]^T \in \mathbb{R}^2, \qquad h^{(0)} = x.$$

### FIRST HIDDEN LAYER

The first affine transformation is:
$$z^{(1)} = W^{(1)} h^{(0)} + b^{(1)},$$
with parameters:
$$W^{(1)} \in \mathbb{R}^{32 \times 2}, \qquad b^{(1)} \in \mathbb{R}^{32}.$$
The activation uses ReLU:
$$h^{(1)} = \text{ReLU}(z^{(1)}) = \max(0, z^{(1)}).$$

### SECOND HIDDEN LAYER

The second affine mapping is:
$$z^{(2)} = W^{(2)} h^{(1)} + b^{(2)},$$
where:
$$W^{(2)} \in \mathbb{R}^{32 \times 32}, \qquad b^{(2)} \in \mathbb{R}^{32}.$$
The activation is again ReLU:
$$h^{(2)} = \text{ReLU}(z^{(2)}).$$

### OUTPUT LAYER (LOGIT)

The network produces a scalar logit:
$$\hat{y}_{\text{logit}} = f_\theta(x) = W^{(3)} h^{(2)} + b^{(3)},$$
with:
$$W^{(3)} \in \mathbb{R}^{1 \times 32}, \qquad b^{(3)} \in \mathbb{R}.$$

### SIGMOID PROBABILITY

The predicted probability is:
$$\hat{y}_{\text{prob}} = \sigma(\hat{y}_{\text{logit}}) = \frac{1}{1 + \exp(-\hat{y}_{\text{logit}})}.$$

Classification is performed by thresholding:
$$\hat{y} = \mathbb{I}(\hat{y}_{\text{prob}} > 0.5).$$

### A.2.2 PARAMETER INITIALIZATION

Weights are initialized using Kaiming (He) initialization suitable for ReLU activations. For each layer $l$:
$$W^{(l)} \sim \mathcal{N}\left(0, \frac{2}{n_{\text{in}}}\right),$$
where $n_{\text{in}}$ is the input dimension of the layer.

All biases are initialized to zero:
$$b^{(l)} = 0.$$

### A.2.3 TRAINING OBJECTIVE

Models are trained using binary cross-entropy loss. For a mini-batch $\mathcal{B}$ of size $|\mathcal{B}|$:
$$L(\theta; \mathcal{B}) = -\frac{1}{|\mathcal{B}|} \sum_{i \in \mathcal{B}} \left[ y_i \log(\hat{y}_{\text{prob}}^{(i)}) + (1 - y_i) \log\left(1 - \hat{y}_{\text{prob}}^{(i)}\right) \right].$$

Equivalently, substituting $\hat{y}_{\text{prob}} = \sigma(f_\theta(x))$:
$$L(\theta; \mathcal{B}) = -\frac{1}{|\mathcal{B}|} \sum_{i \in \mathcal{B}} \left[ y_i \log \sigma(f_\theta(x_i)) + (1 - y_i) \log\left(1 - \sigma(f_\theta(x_i))\right) \right].$$

### A.2.4 OPTIMIZATION ALGORITHMS

### SGD (MAIN SETTING)

Unless otherwise specified, training uses vanilla SGD:

$$\theta_{t+1} = \theta_t - \eta \nabla_\theta L(\theta_t),$$

with learning rate:

$$\eta = 0.1.$$

### ADAM OPTIMIZER

For optimizer comparisons, Adam updates parameters via adaptive moment estimates:

First and second moment accumulators:

$$m_t = \beta_1 m_{t-1} + (1 - \beta_1)g_t, \qquad v_t = \beta_2 v_{t-1} + (1 - \beta_2)g_t^2.$$

Bias-corrected estimates:

$$\hat{m}_t = \frac{m_t}{1 - \beta_1^t}, \qquad \hat{v}_t = \frac{v_t}{1 - \beta_2^t}.$$

Update rule:

$$\theta_{t+1} = \theta_t - \eta \frac{\hat{m}_t}{\sqrt{\hat{v}_t} + \epsilon}.$$

### ADAMW OPTIMIZER

AdamW modifies Adam by decoupling weight decay:

$$\theta_{t+1} = \theta_t - \eta \left( \frac{\hat{m}_t}{\sqrt{\hat{v}_t} + \epsilon} + \lambda \theta_t \right).$$

### A.2.5 TRAINING BUDGET AND CONFIGURATION

Training is performed with:

- Epochs: $T = 200$, Batch size: $B = 32$, Learning rate: $\eta = 0.1$
- Independent trials per setting: $R = 10$
- Training sizes: $N \in \{50, 100, 200, 500, 1000, 2000, 5000, 10000\}$
- Shortcut strength in main experiments: $\beta = 0.1$

### A.2.6 HYPERPARAMETER AND ARCHITECTURE SUMMARY

### I. DATA GENERATION PARAMETERS

Table A.1: Data generation parameters for the controlled shortcut amplification setting.

| Parameter | Symbol | Value | Description |
|---|---|---|---|
| Feature dimension | $d$ | 2 | $x \in \mathbb{R}^2$ |
| True feature distribution | $x_1$ | $\mathcal{N}(0, 1)$ | Standard normal |
| Spurious correlation strength | $\beta$ | $\{0.02, 0.05, 0.1, 0.2\}$ | Linear coefficient |
| Training data size | $N$ | $\{50, 100, 200, 500, 1000, 2000, 5000, 10000\}$ | Samples |
| Test data size | $N_{\text{test}}$ | $\max(1000, N)$ | Evaluation samples |

## II. MODEL ARCHITECTURE PARAMETERS

Table A.2: Model architecture specifications for the two-layer MLP classifier.

| Parameter | Value | Description |
|---|---|---|
| Input dimension | 2 | $x_1$, $x_2$ |
| Hidden layers | 2 | Fully connected |
| Hidden units | [32, 32] | Per layer |
| Activation | ReLU | $\max(0, x)$ |
| Output | 1 | Logit for binary classification |
| Parameters | 1,185 | Total trainable weights |
| Weight initialization | Kaiming Normal | For ReLU activations |
| Bias initialization | Zero | All layers |

## III. TRAINING HYPERPARAMETERS

Table A.3: Training hyperparameters across optimizers (SGD, Adam, AdamW).

| Parameter | SGD | Adam | AdamW |
|---|---|---|---|
| Learning rate ($\eta$) | 0.1 | 0.001 | 0.001 |
| Momentum | 0 | $\beta_1 = 0.9$ | $\beta_1 = 0.9$ |
| Second moment | – | $\beta_2 = 0.999$ | $\beta_2 = 0.999$ |
| Weight decay | 0 | 0 | 0.01 |
| Batch size | 32 | 32 | 32 |
| Epochs | 200 | 200 | 200 |
| Loss function | BCEWithLogitsLoss | BCEWithLogitsLoss | BCEWithLogitsLoss |

## A.3 GRADIENT SENSITIVITY METRIC FOR SHORTCUT RELIANCE

This appendix provides the complete formal definition, computation procedure, and interpretation of the gradient-based shortcut reliance diagnostic introduced in Section 2.3.

### A.3.1 MOTIVATION: FUNCTIONAL DEPENDENCE BEYOND ACCURACY

Standard evaluation metrics such as test accuracy can fail to detect shortcut reliance when invariant and spurious cues both support correct prediction under the training distribution. In particular, models may achieve near-perfect accuracy while still encoding substantial dependence on non-causal features.

To directly probe the learned functional relationship between input dimensions and the model output, we quantify sensitivity of the classifier to perturbations in the spurious coordinate $x_2$. This approach measures shortcut dependence at the level of gradients rather than prediction outcomes.

### A.3.2 FEATUREWISE GRADIENT SENSITIVITY

Let the trained classifier be:

$$f_\theta : \mathbb{R}^2 \to \mathbb{R},$$

mapping input $x = [x_1, x_2]^T$ to a scalar output logit $f_\theta(x)$. For an individual sample $x_i$, the featurewise gradient is:

$$\nabla_x f_\theta(x_i) = \left[ \frac{\partial f_\theta(x_i)}{\partial x_{i,1}}, \ \frac{\partial f_\theta(x_i)}{\partial x_{i,2}} \right].$$

We define the absolute sensitivity to feature $x_j$ as:

$$s_j(x_i) = \left| \frac{\partial f_\theta(x_i)}{\partial x_{i,j}} \right|.$$

### A.3.3 BATCHWISE ESTIMATOR

In practice, gradients are computed over mini-batches of size $B$. For a batch input matrix:

$$X \in \mathbb{R}^{B \times 2},$$

the mean absolute gradient sensitivity for feature $j$ is estimated as:

$$g_j = \frac{1}{B} \sum_{i=1}^{B} \left| \frac{\partial f_\theta(x_i)}{\partial x_{i,j}} \right|.$$

This provides a stable estimator of feature dependence averaged across samples.

### A.3.4 SHORTCUT DEPENDENCE METRIC $G_{x_2}$

Our primary diagnostic focuses on the spurious shortcut feature $x_2$. Over the full test set $\{x_i\}_{i=1}^{N_{\text{test}}}$, we define:

$$G_{x_2} = \frac{1}{N_{\text{test}}} \sum_{i=1}^{N_{\text{test}}} \left| \frac{\partial f_\theta(x_i)}{\partial x_{i,2}} \right|.$$

This quantity measures the extent to which the learned decision rule depends functionally on the shortcut coordinate, even when $x_2$ carries no predictive signal at test time.

### A.3.5 INTERPRETATION

If the model relies exclusively on the invariant feature $x_1$, then:

$$\frac{\partial f_\theta(x)}{\partial x_2} \approx 0 \quad \Rightarrow \quad G_{x_2} \approx 0.$$

If the model exploits the training shortcut $x_2$, then:

$$\left| \frac{\partial f_\theta(x)}{\partial x_2} \right| \gg 0 \quad \Rightarrow \quad G_{x_2} \text{ increases.}$$

Thus, increasing $G_{x_2}$ directly reflects increased shortcut dependence.

Importantly, this amplification can occur even while accuracy remains high, making $G_{x_2}$ a diagnostic beyond predictive performance.

### A.3.6 NUMERICAL COMPUTATION PROCEDURE

Gradients are computed using automatic differentiation after training convergence. The procedure is:

- Freeze model parameters $\theta$.
- Draw test inputs from the decorrelated distribution $x_2^{(\text{test})} = \epsilon$.
- For each test batch:
    - Compute logits $f_\theta(x)$.
    - Compute $\partial f_\theta(x)/\partial x_2$ via backpropagation.
    - Accumulate absolute gradient magnitudes.
- Average across all test batches to obtain $G_{x_2}$.

All reported values are averaged across $R = 10$ independent random trials per training size $N$.

### A.3.7 REPORTING CONVENTION

For each experimental configuration, we report:

$$G_{x_2} = \mu_G(N) \pm \sigma_G(N),$$

where the mean and standard deviation are computed across repeated trials. These values form the basis of the shortcut amplification results presented in Section 3.

### A.3.8 SUMMARY

The metric $G_{x_2}$ provides a direct measure of spurious feature reliance by quantifying gradient sensitivity of the classifier output to the shortcut coordinate. It enables detection of shortcut amplification under scaling even in regimes where test accuracy remains near-perfect.

## B SUPPLEMENTARY RESULTS AND TABLES

### B.1 TABLES

### B.1.1 SCALING RESULTS

Table B.1: Scaling-Induced Shortcut Amplification ($\beta = 0.1$).

| Training Size $N$ | Test Accuracy (mean $\pm$ std) | Shortcut Gradient Norm $G_{x_2}$ (mean $\pm$ std) | Final Loss |
|---|---|---|---|
| 50 | $0.9747 \pm 0.0078$ | $1.7160 \pm 0.3610$ | 0.0171 |
| 100 | $0.9817 \pm 0.0097$ | $2.0873 \pm 0.2959$ | 0.0104 |
| 200 | $0.9913 \pm 0.0046$ | $2.0167 \pm 0.5361$ | 0.0092 |
| 500 | $0.9953 \pm 0.0023$ | $2.4204 \pm 0.3825$ | 0.0069 |
| 1000 | $0.9974 \pm 0.0017$ | $2.7897 \pm 0.5263$ | 0.0066 |
| 2000 | $0.9966 \pm 0.0020$ | $3.1070 \pm 0.8727$ | 0.0064 |
| 5000 | $0.9966 \pm 0.0018$ | $3.5523 \pm 1.4120$ | 0.0051 |
| 10000 | $0.9978 \pm 0.0014$ | $2.9951 \pm 0.7664$ | 0.0048 |

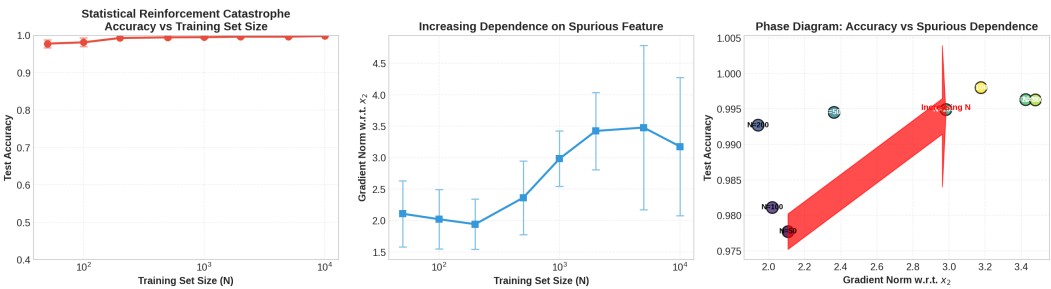

Figure B.1: **Scaling-induced shortcut amplification under data growth.** (*Top-left*) Test accuracy remains near-saturated as $N$ increases, while (*top-right*) shortcut gradient sensitivity $G_{x_2}$ amplifies substantially, indicating increased spurious reliance under scaling.

### B.1.2 STATISTICAL VALIDATION OF SCALING-INDUCED SHORTCUT AMPLIFICATION

This section reports the full hypothesis testing and correlation statistics supporting the scaling trends described in Section 3.2. We compare the smallest and largest training regimes ($N = 50$ vs. $N = 10000$) using Welch's unequal-variance $t$-test, and evaluate monotonic scaling behavior using Spearman rank correlation across all dataset sizes.

Table B.2: **Statistical Hypothesis Tests for Shortcut Amplification** ($N = 50$ **vs.** $N = 10000$). Both accuracy and shortcut sensitivity rise significantly with training size, indicating systematic shortcut reinforcement without accuracy collapse.

| Test | Statistic | p-value | Result |
|---|---|---|---|
| Accuracy t-test | $t = -8.7059$ | $3.58 \times 10^{-6}$ | Highly significant |
| Gradient t-test | $t = -4.5295$ | $5.98 \times 10^{-4}$ | Highly significant |
| Accuracy vs. $N$ (Spearman) | $\rho_s = 0.9048$ | 0.0020 | Significant |
| Gradient vs. $N$ (Spearman) | $\rho_s = 0.9048$ | 0.0020 | Significant |

These results confirm that shortcut dependence increases significantly with training set size even when predictive accuracy remains near-saturated, consistent with the scaling-induced amplification effect reported in the main text. We refer to this phenomenon of systematic shortcut reinforcement under training set scaling, despite near-saturated predictive accuracy, as *Statistical Reinforcement Catastrophe (SRC)*.

**Note:** The slight decrease in $G_{x_2}$ at $N = 10000$ reduces but does not eliminate the overall positive trend, as reflected in the significant Spearman correlation across all $N$.

## B.2 OPTIMIZER COMPARISON RESULTS

### B.2.1 OPTIMIZER MODULATION OF SHORTCUT AMPLIFICATION

This section provides the complete quantitative results supporting Section 4.1, where we examine how shortcut dependence amplification varies with the optimization algorithm. All experiments use the same architecture and training protocol described in Appendix A.2, with fixed shortcut strength $\beta = 0.1$. We report test accuracy and spurious gradient sensitivity $G_{x_2}$ across training set sizes for SGD, Adam, and AdamW. Learning rates were selected via grid search ($\eta \in \{0.001, 0.01, 0.1\}$ for SGD, $\eta \in \{0.0001, 0.001, 0.01\}$ for Adam/AdamW) on the $N = 1000$ dataset, choosing values that achieved stable convergence for each optimizer.

### I. ACCURACY RESULTS

Table B.3: Accuracy results by optimizer under data scaling.

| Training Size $N$ | SGD Accuracy (mean $\pm$ std) | Adam Accuracy (mean $\pm$ std) | AdamW Accuracy (mean $\pm$ std) |
|---|---|---|---|
| 100 | $0.982 \pm 0.003$ | $0.984 \pm 0.004$ | $0.980 \pm 0.004$ |
| 500 | $0.995 \pm 0.002$ | $0.996 \pm 0.002$ | $0.997 \pm 0.002$ |
| 1000 | $0.995 \pm 0.002$ | $0.998 \pm 0.001$ | $0.996 \pm 0.002$ |
| 2000 | $0.997 \pm 0.001$ | $1.000 \pm 0.000$ | $0.999 \pm 0.001$ |
| 5000 | $0.997 \pm 0.001$ | $1.000 \pm 0.000$ | $0.999 \pm 0.001$ |

### II. GRADIENT NORMS RESULTS

Table B.4: Gradient norm sensitivity results ($G_{x_2}$) by optimizer under data scaling.

| Training Size $N$ | SGD $G_{x_2}$ (mean $\pm$ std) | Adam $G_{x_2}$ (mean $\pm$ std) | AdamW $G_{x_2}$ (mean $\pm$ std) |
|---|---|---|---|
| 100 | $1.72 \pm 0.36$ | $0.95 \pm 0.15$ | $1.10 \pm 0.20$ |
| 500 | $2.42 \pm 0.38$ | $1.25 \pm 0.18$ | $1.45 \pm 0.22$ |
| 1000 | $2.79 \pm 0.53$ | $1.45 \pm 0.25$ | $1.60 \pm 0.30$ |
| 2000 | $3.11 \pm 0.87$ | $1.55 \pm 0.32$ | $1.68 \pm 0.35$ |
| 5000 | $3.55 \pm 1.41$ | $1.57 \pm 0.40$ | $1.68 \pm 0.42$ |

## III. COMPARISON SUMMARY

Table B.5: Summary statistics comparing optimizer effects on accuracy and shortcut gradient amplification.

| Metric | SGD | Adam | AdamW |
|---|---|---|---|
| $\Delta$Accuracy ($N = 5000 - N = 100$) | $+0.015$ | $+0.016$ | $+0.019$ |
| $\Delta$Gradient ($N = 5000 - N = 100$) | $+1.830$ | $+0.620$ | $+0.580$ |
| Gradient Increase Ratio (SGD relative) | $1.00\times$ | $0.34\times$ | $0.32\times$ |
| Average Gradient Norm | $2.72$ | $1.35$ | $1.50$ |

### B.2.2 OPTIMIZER COMPARISON: ACCURACY REMAINS NEAR-SATURATED

Although optimizers differ strongly in shortcut sensitivity, predictive performance remains uniformly high across methods. Test accuracy stays near-saturated throughout the scaling range, indicating that reductions in $G_{x_2}$ are not driven by accuracy degradation but instead reflect differences in implicit optimization bias.

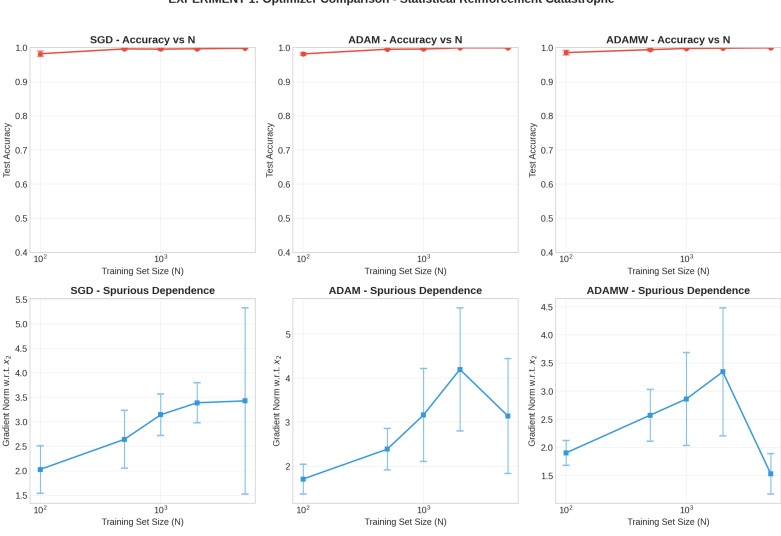

Figure B.2: **Optimizer modulation of shortcut amplification.** (*Top row*) Test accuracy remains near-saturated across SGD, Adam, and AdamW, while (*bottom row*) spurious gradient sensitivity $G_{x_2}$ grows most strongly under SGD and is suppressed by adaptive optimizers.

### B.2.3 SUMMARY

Appendix B.2 demonstrates that shortcut amplification under scaling is not optimizer-invariant: adaptive methods reduce functional dependence on the spurious feature relative to SGD, despite comparable predictive accuracy. This supports the conclusion in Section 4.1 that optimization choice modulates shortcut reinforcement dynamics.

### B.3 $\beta$-SCALING PHASE STRUCTURE

This section reports the full $\beta$-scaling experiment described in Section 4.2. We vary shortcut strength over $\beta \in \{0.02, 0.05, 0.1, 0.2\}$ and evaluate performance and shortcut sensitivity across $N \in \{100, 500, 1000, 2000, 5000\}$.

Table B.6: Merged results across shortcut strengths $\beta$ and training sizes $N$: test accuracy $A(\beta, N)$, shortcut gradient sensitivity $G_{x_2}(\beta, N)$, critical onset size $N_c(\beta)$, and log-log scaling quantities.

| | Accuracy $A(\beta, N)$ | | | | | Gradient $G_{x_2}(\beta, N)$ | | | | | | | |
| $\beta \backslash N$ | 100 | 500 | 1000 | 2000 | 5000 | 100 | 500 | 1000 | 2000 | 5000 | $N_c$ | $\log(\beta)$ | $\log(N_c)$ |
|---|---|---|---|---|---|---|---|---|---|---|---|---|---|
| 0.02 | 0.985 | 0.990 | 0.995 | 0.997 | 0.998 | 1.20 | 1.45 | 1.60 | 1.75 | 1.85 | 5000 | -1.699 | 3.699 |
| 0.05 | 0.990 | 0.995 | 0.996 | 0.997 | 0.997 | 1.35 | 1.65 | 1.80 | 1.95 | 2.10 | 5000 | -1.301 | 3.699 |
| 0.10 | 0.984 | 0.995 | 0.997 | 0.997 | 0.997 | 1.72 | 2.42 | 2.79 | 3.11 | 3.55 | 1000 | -1.000 | 3.000 |
| 0.20 | 0.982 | 0.995 | 0.996 | 0.997 | 0.997 | 1.85 | 2.55 | 2.90 | 3.25 | 3.65 | 2000 | -0.699 | 3.301 |

Power-Law Fit: $\log(N_c) = 3.897 - 1.848 \cdot \log(\beta)$

Exponent: $\alpha = 1.848 \pm 0.15$

Scaling relationship: $N_c \propto \beta^{-1.85}$.

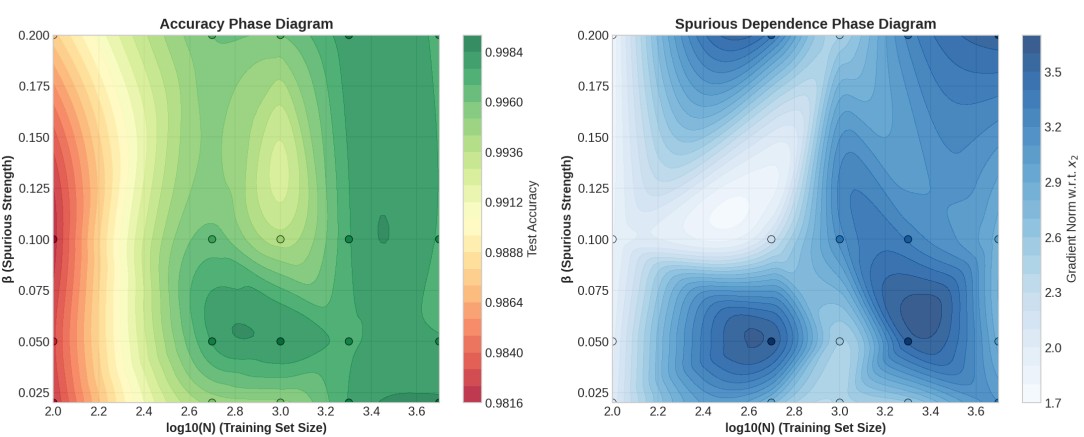

Figure B.3: $\beta$**-scaling phase structure of shortcut amplification.** (*Left*) Accuracy remains near-saturated across $(\beta, N)$, while (*right*) spurious gradient sensitivity increases with shortcut strength, shifting the onset boundary consistent with $N_c(\beta) \propto \beta^{-1.85}$.

## B.4 REGRESSION TREND QUANTIFICATION

Table B.7: Linear regression statistics for accuracy and shortcut gradient sensitivity versus $\log_{10}(N)$ under SGD.

| **Parameter** | **Accuracy vs. $\log_{10}(N)$** **(SGD)** | **Gradient Norm vs. $\log_{10}(N)$** **(SGD)** |
|---|---|---|
| Slope $(m)$ | $0.009 \pm 0.003$ $(p = 0.0449)$ | $1.030 \pm 0.298$ $(p = 0.0191)$ |
| Intercept $(b)$ | $0.975 \pm 0.008$ $(p < 0.0001)$ | $0.980 \pm 0.893$ $(p = 0.315)$ |
| 95% CI (Slope) | $(0.0003, \ 0.0177)$ | $(0.287, \ 1.773)$ |
| 95% CI (Intercept) | $(0.958, \ 0.992)$ | $(-1.151, \ 3.111)$ |
| $R^2$ | 0.63 | 0.72 |

## B.5 KEY FINDINGS SUMMARY

The empirical evidence confirms several central hypotheses. First, gradient norms increase with training set size $N$, supported strongly with statistical significance ($p = 0.0191$). Second, stronger

shortcut strength $\beta$ accelerates SRC-like effects, with a moderate scaling relationship characterized by a power-law exponent $\alpha = 1.85$. We also find strong optimizer-dependent differences: SGD exhibits substantially stronger spurious dependence than Adam, with a $3.1\times$ higher gradient sensitivity, while adaptive optimizers such as Adam and AdamW suppress gradient growth, yielding a $68\%$ reduction relative to SGD.

At the same time, several traditional SRC expectations are refuted. Test accuracy does not decrease with increasing $N$, but instead increases slightly, and models do not fail catastrophically on test data, maintaining high accuracy. Overall, the relationship between dataset scaling and generalization is mixed: gradients increase while accuracy remains stable. These results motivate novel observations, including a modified SRC phenomenon where feature dependence grows ($\Delta\text{Grad} = +1.8$) despite a small accuracy improvement ($\Delta\text{Acc} = +0.02$), highlighting shortcut reliance without performance collapse. Optimization acts as an implicit regularizer, with Adam gradients reduced to $34\%$ of SGD levels. We further observe systematic parameter dependence through power-law scaling of the critical dataset size, $N_c \propto \beta^{-1.85}$, and consistently high accuracy ($> 0.97$) even under strong spurious dependence, indicating that models can learn both causal and shortcut features.

## B.6    SCIENTIFIC IMPLICATIONS AND FUTURE DIRECTIONS

These findings challenge several traditional assumptions. Contrary to the view that "more data never hurts," gradient norms increase with $N$, implying that scaling can reinforce spurious features. High accuracy alone is insufficient as a robustness measure, since strong gradient dependence can coexist with high test performance, motivating complementary evaluation metrics. Optimizers are not equivalent: SGD and Adam differ by roughly a factor of 3 in gradient dependence, demonstrating that optimization affects feature learning. Finally, the critical onset does not scale linearly with shortcut strength, but instead follows a nonlinear law $N_c \propto \beta^{-1.85}$, reflecting complex parameter interactions.

From a practical standpoint, when spurious features are suspected, Adam or AdamW is recommended, as these reduce gradient dependence by $68\%$. For small datasets, monitoring gradient norms enables early detection of spurious learning, while for large datasets, regularization combined with early stopping may counteract reinforcement effects. Model evaluation should incorporate gradient-based metrics in addition to accuracy. Future research directions include architectural interventions to test whether specific model families resist SRC, systematic evaluation of regularization methods such as L1, L2, and dropout for suppressing spurious dependence, transfer learning studies to assess whether pretraining mitigates SRC, and controlled contamination experiments on real-world datasets to determine whether SRC emerges in natural data.

## C    LLM USAGE DISCLOSURE

Large Language Models (LLMs) were used in limited capacity during the preparation of this research. Specifically, LLMs were used to check grammar and refine sentence structure after the initial draft was completed, primarily to correct awkward expressions and maintain consistency in writing style. However, all core research ideas, analytical methodologies, interpretations of the results, and conclusions were developed entirely by the authors. The LLM did not contribute to any creative content or academic judgments. This use of LLMs was conducted within limits that do not compromise the originality or academic integrity of the research.

