# OpenReview forum: "When Data Amplifies Shortcuts: Gradient-Flow Evidence of Spurious Feature Reinforcement"
_ICLR.cc/2026/Workshop/Sci4DL — Submitted to Sci4DL 2026_

### Official Review · Reviewer_ThwU · 2026-02-25

**Fit:** 3
**Significance:** 3
**Confidence:** 2

**Summary:**

This paper studies how increasing dataset size can amplify shortcut learning in modern machine learning models. Contrary to the common assumption that more data improves generalization, the authors show that when datasets contain spurious correlations, scaling up data can actually reinforce reliance on these shortcuts rather than mitigate them. Through controlled experiments, the paper demonstrates that models increasingly exploit easily learnable but non-causal features as dataset size grows. The authors analyze this phenomenon across different settings and highlight conditions under which scaling fails to recover the true underlying signal. The work raises concerns about data scaling as a universal solution and calls for more robust training strategies.

**Strengths:**

Timely and important question: Challenges the widely held belief that “more data fixes everything,” which is highly relevant in the era of large-scale training.

Clear empirical evidence: The paper provides controlled experiments that isolate shortcut features and convincingly show their amplification with scale.

Conceptual clarity: The framing of shortcut learning as a competition between signal and spurious features is intuitive and well-articulated.

Practical implications: Highlights real risks in large-scale training pipelines, especially in settings with latent confounders.

Broad relevance: The phenomenon is likely applicable beyond the specific experimental setups, including vision and language models.

**Suggestions:**

Limited theoretical depth: While the empirical findings are compelling, the paper lacks a strong theoretical framework explaining why scaling favors shortcuts in general settings.

Synthetic or controlled setups: Much of the evidence appears to rely on constructed datasets with explicit spurious correlations; it is less clear how strongly the effect manifests in fully natural datasets.

Mitigation strategies underexplored: The paper focuses on diagnosing the problem but provides limited guidance on how to address it (e.g., reweighting, invariance methods, data augmentation).

Scope of architectures: It is unclear whether the observed behavior is consistent across a wide range of model classes or specific to the ones tested.

Quantification of “shortcut reliance”: The metrics used to measure shortcut usage could be discussed more rigorously or compared to alternative definitions.

---

### Official Review · Reviewer_t5g5 · 2026-02-26

**Fit:** 3
**Significance:** 2
**Confidence:** 3

**Summary:**

The paper studies shortcut learning under a controlled synthetic distribution shift with two features: an invariant causal feature $x_1$ that determines the label, and a spurious training shortcut $x_2$ correlated with the label only during training. The authors studied a binary classification task, how shortcut reliance evolves with dataset scaling.

**Strengths:**

1. The authors introduce a direct functional measure of shortcut reliance, spurious-feature gradient sensitivity, showing shortcut dependence can grow even when test accuracy is near-saturated with dataset scaling.

2. The synthetic two-feature construction (causal $x_1$, spurious $x_2$) with an explicit train–test decorrelation makes the shortcut mechanism easy to isolate and interpret.

**Suggestions:**

1. The optimizer result is interesting. Adam/AdamW substantially reduces spurious gradient sensitivity relative to SGD under the authors’ synthetic setup. However, this appears primarily empirical, and the underlying mechanism is not isolated, so additional investigation across settings (architectures, hyperparameter matching, and real datasets) would strengthen the claim.

2. The related-work discussion would be stronger if it positioned the observed shortcut amplification and optimizer effects relative to prior theory on simplicity bias[1] and gradient starvation[2], which are not currently cited or discussed.

[1] The Pitfalls of Simplicity Bias in Neural Networks, Harshay Shah and Kaustav Tamuly and Aditi Raghunathan and Prateek Jain and Praneeth Netrapalli, NIPS'20

[2] Gradient starvation: a learning proclivity in neural networks, Pezeshki, Mohammad and Kaba, S\'{e}kou-Oumar and Bengio, Yoshua and Courville, Aaron and Precup, Doina and Lajoie, Guillaume, NIPS' 21

---

### Official Review · Reviewer_nfyD · 2026-02-26

**Fit:** 2
**Significance:** 1
**Confidence:** 3

**Summary:**

The paper’s central claim is that increasing the amount of training data can lead to greater reliance on spurious features, as measured by a proposed notion of gradient sensitivity, while predictive accuracy remains unchanged. The empirical analysis is conducted in a two-dimensional synthetic setting where one dimension causally determines the label and the other is spuriously correlated with it. The authors further present evidence suggesting that adaptive optimizers exhibit lower dependence on the spurious feature.

However, the paper contains a fundamental flaw that undermines its conclusions. Gradient sensitivity is defined via the sensitivity of the logits to the spurious input dimension. Because the training procedure does not include weight decay, continued optimization naturally increases the magnitude of the model weights, which in turn increases sensitivity to both input dimensions. Since the batch size is fixed and all models are trained for 200 epochs regardless of dataset size, larger datasets result in more parameter updates and therefore larger logit scales and sensitivities. Consequently, the reported trend with data size is confounded by optimization dynamics, and the results are misinterpreted.

The observed lower sensitivity for Adam and AdamW can be explained by differences in effective learning rates rather than by adaptivity per se. As reported in Table A.3, these optimizers use substantially smaller learning rates than SGD, leading to slower growth in logit magnitudes. In addition, AdamW applies weight decay, introducing further regularization. Therefore, the findings do not isolate the effect of adaptivity. To properly evaluate whether adaptive methods reduce reliance on spurious features, the experiments should control for learning rate schedules, number of optimization steps, and regularization so that comparisons are fair along these dimensions.

**Strengths:**

The presentation is clear and easy to follow. The authors provide sufficient experimental details, enabling a well-grounded assessment of the work. The problem the paper addresses, namely the relationship between dataset size and reliance on spurious correlations, is both relevant and important. The paper also examines the role of optimization algorithms in robustness to spurious correlations, a topic that has recently attracted considerable attention in the community, as reflected in several recent works [1, 2, 3].

[1] Vasudeva et al., How Muon’s Spectral Design Benefits Generalization: A Study on Imbalanced Data, ICLR 2026.
[2] Mirzaei et al., On the Role of Implicit Regularization of Stochastic Gradient Descent in Group Robustness, ICLR 2026.
[3] Ghaznavi et al., On the Potential of the Four-Point Model for Studying the Role of Optimization in Robustness to Spurious Correlations, 17th Annual Workshop on Optimization for Machine Learning at NeurIPS 2025.

**Suggestions:**

The authors should design experiments that are fair with respect to the number of optimization iterations when supporting their claim. In particular, they need to demonstrate that the observed phenomenon is not merely a consequence of increasing logit magnitudes during training, but persists even when the scale of the logits is properly controlled.

---

### Meta-Review · Area_Chair_rj4j · 2026-02-28

**Recommendation:** Reject

**Metareview:**

This paper investigates shortcut learning and its interaction with dataset size. However, as noted by reviewer nfyD, there are critical issues with the methodology. In addition, the paper is out of scope for the workshop, as it does not conduct experiments on real-world datasets. I recommend rejection.

---

### Decision · Program_Chairs · 2026-03-02

Reject